# RESTORE: Robust intEnSiTy nORmalization mEthod for multiplexed imaging

Young Hwan Chang [1,2,3✉], Koei Chin[2,3], Guillaume Thibault[2], Jennifer Eng[2], Erik Burlingame[1,2] & Joe W. Gray[2,3✉]

Recent advances in multiplexed imaging technologies promise to improve the understanding of the functional states of individual cells and the interactions between the cells in tissues. This often requires compilation of results from multiple samples. However, quantitative integration of information between samples is complicated by variations in staining intensity and background fluorescence that obscure biological variations. Failure to remove these unwanted artifacts will complicate downstream analysis and diminish the value of multiplexed imaging for clinical applications. Here, to compensate for unwanted variations, we automatically identify negative control cells for each marker within the same tissue and use their expression levels to infer background signal level. The intensity profile is normalized by the inferred level of the negative control cells to remove between-sample variation. Using a tissue microarray data and a pair of longitudinal biopsy samples, we demonstrated that the proposed approach can remove unwanted variations effectively and shows robust performance.

[1] Computational Biology Program, Oregon Health & Science University, Portland, OR 97201, USA. [2] Department of Biomedical Engineering, OHSU Center for Spatial Systems Biomedicine, Oregon Health & Science University, Portland, OR 97201, USA. [3] Knight Cancer Institute, Oregon Health & Science University, Portland, OR 97201, USA. ✉email: chanyo@ohsu.edu; grayjo@ohsu.edu

Recent developments in multiplexed staining and imaging such as cyclic immunofluorescence (CyCIF)[1–3], multiplexed immunohistochemistry (IHC)[4], CODEX[5], and other multiplexed imaging methods have greatly expanded the palette that researchers and pathologists can use to visualize and analyze tissue sections, allowing deep in situ assessment of the tumor microenvironment complexities[6–11]. Multiplexed analyses have the advantages of both highly multiplexed detection and retention of morphological context at the level of single-cell and subcellular compartments. However, integration of information from multiple samples is challenging due to the lack of normalization procedures that can correct for technical variations in staining intensities between samples that result from variations in fixation, antibody concentration, etc. so that biological differences between samples can be accurately assessed[6,12,13].

At present, the most common approach for the quantitative assessment of images of IHC- and IF-labeled material is an analysis technique commonly referred to as "gating" or "binary thresholding" based on single-cell features. Essentially, a particular pixel intensity level (the threshold) is manually defined and then used to demarcate what is considered to be "signal" (the immuno-labeled material of interest) and "noise" (non-specific material attributable to the immuno-labeling process). This manual thresholding procedure can only provide genuinely valid results if one adjusts the threshold cut-point for each individual sample to deal with such intensity variations. This approach is often used during analysis of multiplex IHC data[4]. However, manual thresholding is subjective and cannot be scalable by its nature. Our recent analyses of images acquired during CyCIF analyses of tissue microarrays (TMAs) showed that tissue-to-tissue variation in autofluorescence and/or nonspecific immunofluorescent staining required manual setting of individual thresholds for each tissue and marker, creating a bottleneck and introducing bias. As an extreme example, we observed in some cases that the intensity value of negative cells in one tissue was higher than the values of positive cells in a separate sample. In this case, setting a global threshold was not possible. This extreme variation also precluded taking full advantage of the quantitative nature of the fluorescence images.

Recently, unsupervised clustering approaches[14–17] have been adopted for identification of different cell types from a continuous intensity distribution, instead of binarization. However, the sample-to-sample intensity variations due to the technical artifact throughout the procedure cause cells to cluster by samples or batches, instead of their cell types. For the above example we mentioned, applying unsupervised clustering without compensating for intensity variation will yield a mixture of positive and negative cells from two samples due to batch effects. Therefore, the corrections of the unwanted intensity variations due to technical artifact and batch-effect within a group of samples are required as pre-processing steps for unsupervised clustering approaches to identify cell types. Quantile normalization (QN) in which intensity measurements that encompass values between 1st and the 99th percentile are aligned, is often used as a pre-processing step[2] to normalize intra-sample variations. However, if samples contain few or no positive cells for a certain marker, different cell populations, or different intensity distributions, QN may cause confounding variations by changing overall intensity profiles.

The general problem is that the intensity features produced by multiplexed immunostaining have different intensity variations across markers, tissues, batches, etc. As an example, Fig. 1 shows intensity variation from three adjacent sections (considered as *almost* technical replicates since each section is acquired with 5 µm thickness difference) where these unwanted variations may be caused by technical issues such as batch effect, exposure time,

protocols or tissue preparation. Note that in our previous study[24], we were able to register one section (Hematoxylin and Eosin stain) to the other section (IF imaging) based on nuclear staining (hematoxylin-stain and DAPI) where they differ by 5 µm thickness, which guarantees little variation in cell population within 5 µm difference. These sections were stained on separate days using CyCIF for 20 proteins and phosphoproteins identifying tumor, immune and stromal cells and functional states (the downloadable data sets carry 20 biomarkers (except DAPI) and additional 17 markers shown in Supplementary Table 1 are validated in the protocol). Individual cells in the multiplex images were segmented using watershed segmentation followed by morphological operation and staining intensities were calculated for each segmented cell[3,18]. Figure 1 shows t-SNE projection based on mean intensity profiles, where each dot represents a single-cell feature and red, green, and blue color indicate sections 1, 2, and 3, respectively. The top region of t-SNE[25] shows cancer cell clusters from each section but due to intensity variation of cytokeratin (CK) markers, it shows batch effect (i.e., red, green, and blue clusters). In contrast to cancer cell types, the bottom region of t-SNE projection represents immune cell types showing a more uniformly distributed pattern between samples, which we expect to see in the ideal setting, i.e., if there is no intensity variation across three adjacent sections.

In IHC, the antibody validation reference is performed with control tissues known to contain the antigen of interest detected by an identical staining method. For example, the "sausage technique" has been used where the entire sample has been proposed as a reference control standard[19,20]. It is recommended that validation studies should be carried out on multi-tissue control blocks containing both known-positive and known-negative normal and tumor tissues[21]. If we consider only a few markers in typical IHC or IF staining, it is feasible to include control tissue samples in the staining/imaging process for individual markers and address antibody validation or staining variation by determining an appropriate transformation or normalization for each marker. Unfortunately, it is not practical to add control tissue samples for reference of individual CyCIF channels since there exist tens of markers (>40) that may require more control tissue samples than the test sample to cover various cell types and functional states. In addition, even though we could normalize features based on reference samples, there still exist other uncontrollable factors causing intensity variations such as tissue fixation, processing in a different way than the test tissue, and antibody lot-to-lot variations especially for large cohort study.

In parallel, computational approaches have been proposed to adjust for unwanted variation and can be divided into two broad categories: (1) global adjustment and (2) an application-specific method[22]. As an example, QN, in which intensity measurements that encompass values between 1st and the 99th percentile are aligned, generally regarded as a self-contained step that plays no role in the downstream analysis of the data, belongs to the first category. In the second category, we find methods that incorporate adjustment into the main analysis of interest. For instance, the batch effects can be handled by explicitly adding "batch terms" to a linear model. Other linear model-based methods such as factor analysis attempt to infer the unwanted variation from the data and then adjust for it. However, these models require technical replicates or a priori information to identify batch effects, which is not feasible in our case, especially because tissue biopsies are precious and are often difficult to obtain in sufficient quantities. In addition, for needle biopsy samples, it is difficult to consider individual region of interests as technical replicates due to the intrinsic heterogeneity.

We propose RESTORE for image quantification in a multiplexed imaging platform to address these issues. A key feature in

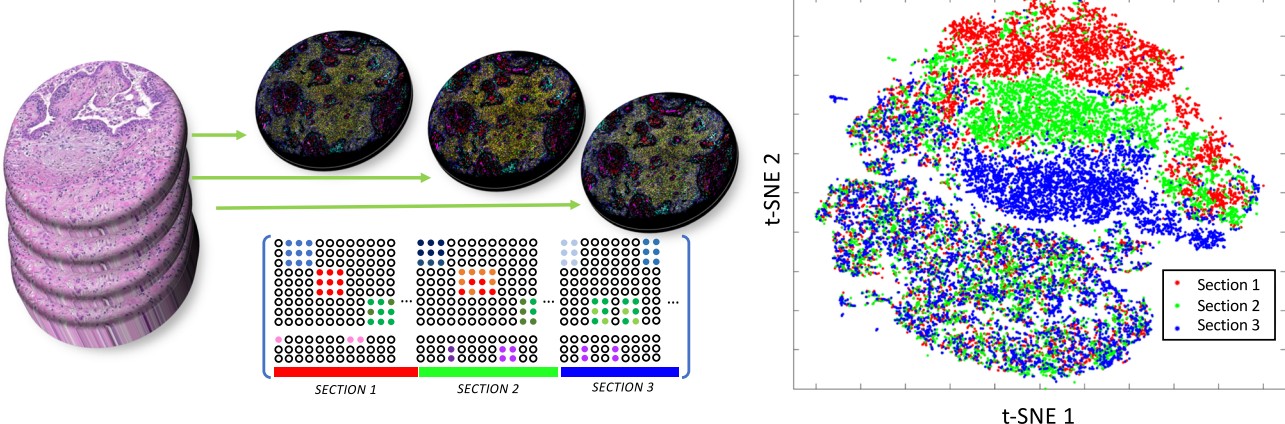

**Fig. 1 An illustration of immunostaining intensity variation.** Multiplexed immunostaining intensity varies across samples; intensity profiles from three adjacent sections (5 μm) show intensity variations. t-SNE embedding of individual single-cell intensity features from three adjacent sections show uniformly distributed in immune cell types (bottom half) but clustered group in cancer cell type (top half) due to the intensity variation in cytokeratin (CK) markers.

RESTORE is the recognition that some cell types defined by reference markers in the multiplex can be safely assumed to have background or noise levels (herein, we consider intensity level of negative cells defined by mutually exclusive marker pairs as background or simply noise level, i.e., interchangeably used) for other markers in the multiplex. For example, immune cells can be assumed to have background or noise levels of cancer-associated cytokeratins and vice versa. RESTORE uses these orthogonal staining patterns to set negative staining thresholds as a guide to further image intensity normalization. We then make the simplifying assumption that signals in the positive channels can be normalized by dividing by the inferred background level. This assumption is inaccurate for low signal levels but becomes increasingly accurate for high signals based on our simulation study (Section: Intensity normalization by inferring auto-fluorescence signal). We demonstrate the feasibility of the proposed approach for intensity normalization of multiplexed image data for robust analysis and comparison of tissue samples during analysis of a data set comprised of CyCIF analyses of three adjacent sections cut from a TMA comprised of 59 cores from diverse breast cancers (TMA was composed of 75 breast cancer tissues (two cores each) and 59 tissues were analyzed in this study based on tumor content and overall quality of staining) and stained for 20 proteins and phosphoproteins selected to identify diverse tumor and stromal cells and functional status.

## Results

One could consider two approaches to evaluate the proposed approach: (1) use cell classification based on manual gating as a ground truth, or (2) use technical replicates as a ground truth. The former approach needs manual gating for cell-type classification, but it might be subjective and time consuming to classify various cell types from many TMAs (177 total, 59 × 3 adjacent sections as shown in Fig. 2a). For the latter one, we can use 59 TMAs that have three adjacent tissue sections. Since each TMA sample has three adjacent tissue samples, we could consider them as *almost* technical replicates where we expect to see a similar cell type component in their population. For instance, if there is no intensity variation across these three adjacent tissue samples, no matter where we draw threshold line or gate, the population of those cell types should be similar across three adjacent tissue samples. Thus, we measure the cell component as a metric and evaluate how the proposed approach compensates intensity variation. We also compare the correlation coefficient of cell

composition between three adjacent sections with and without using the proposed normalization technique. Finally, we illustrate a clinical use case by applying the proposed approach to the study with longitudinal biopsies.

**Application with TMAs analysis using 3 adjacent tissue sections.** We compare the correlation coefficient of individual cell populations of three adjacent TMAs across 59 samples where $r_{ij}$ represents correlation coefficient of cell population between the $i$th and the $j$th section as shown in Fig. 2a. We define cell population by two approaches: (1) calculating positive cell count for individual markers by inferring background signal and (2) using an unsupervised clustering approach by changing the number of clusters.

**Cell population comparison by inferring background signal.** First, we count positive cell population across all the CyCIF markers by inferring background signal. In order to illustrate intensity variations across three adjacent sections, we infer background signal in two different approaches. For a local approach, we infer background signal for an individual adjacent section, and for a global approach, we combine intensity features across three adjacent sections and infer background signal. If there is no intensity variation, we expect the local and global approaches to show similar results. With a local approach, we observe high correlation of cell composition across three adjacent TMAs as shown in Fig. 2b. However, with a global approach, the correlation of cell composition across three adjacent sections is poor, due to the intensity variation across samples as shown in Fig. 2c. We note that in Fig. 2b (the proposed result), we found that only three cores in TMAs show poor correlation coefficient due to the technical artifacts such as segmentation or staining issues.

Second, we compare the correlation of cell populations across individual markers ($n = 18$) as shown in Fig. 3. Similar to the previous result, the local approach shows better correlation, which confirms that the proposed method provides a robust cell classification result by inferring background signal from the negative control group and compensating intensity variation.

Lastly, we use the coefficient of variation ($c_v = \sigma/\mu$, known as relative standard deviation) of positive cell count based on the inferred threshold of background signal where $\mu$ and $\sigma$ represent mean and standard deviation of positive cell counts across three

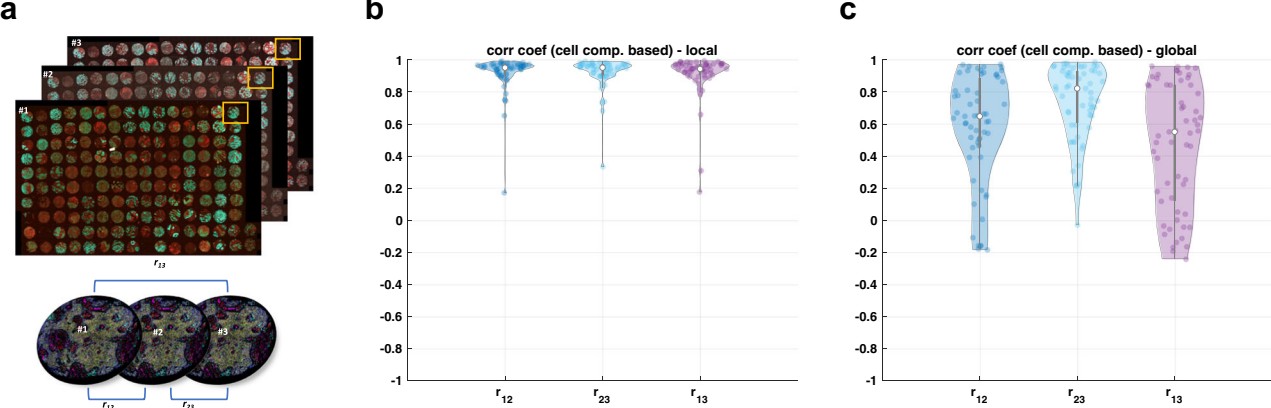

**Fig. 2 Cell component comparison with three adjacent TMA sections.** Example of three adjacent TMA sections (**a** $n = 59$ with three adjacent sections) and correlation coefficient based on group component using unsupervised clustering across these samples; **b**, **c** represent correlation coefficient with and without the proposed approach (RESTORE), respectively, where each dot represents single TMA core. Note that in the result of the proposed approach, three TMA cores show poor correlation coefficient (below 0.8) but we confirm that those three cores show technical artifacts such as segmentation or tissue loss.

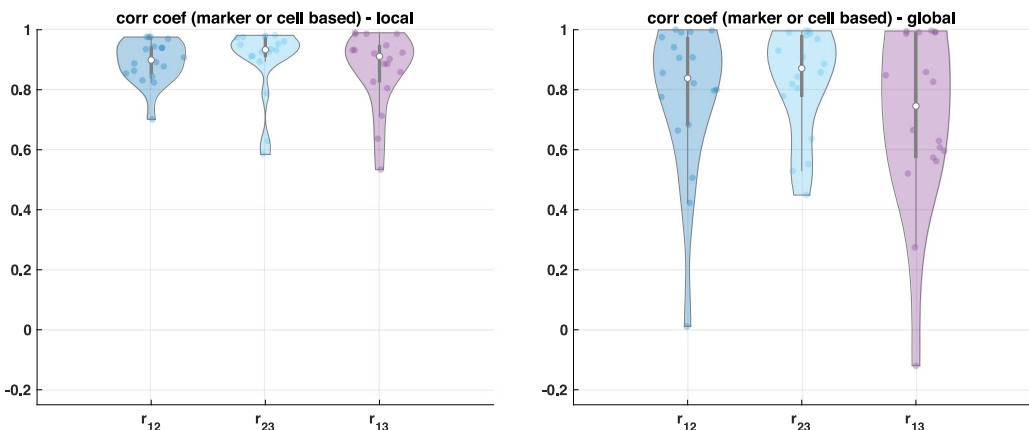

**Fig. 3 Cell component comparison across individual CyCIF markers.** Correlation coefficient based on cell component across individual markers ($n = 18$).

adjacent TMAs, respectively. The coefficient of variation is a standardized measure of dispersion of a frequency distribution, which is commonly used in analytical chemistry to express the precision and repeatability of an assay. We use coefficient of variation to access the positive cell number across individual markers. Figure 4 shows the extent of variability of positive cell count in relation to the mean of population across individual marker. Since there exists intensity variation across CyCIF intensity features, the coefficient of variation of the global approach shows large variation compared with the coefficient of variation of the local approach. For CD45 marker, although the global approach shows slightly better performance, we note that a scale of the value is quite small compared with other $c_v$ for different markers. Similarly, for Ki67 marker, we do not see much difference between the local and global approach.

**Cell population comparison with and without the proposed intensity normalization by using unsupervised clustering approach.** We apply an unsupervised clustering algorithm (k-means) with and without the proposed intensity normalization method on the three adjacent sections to define cell types. If the proposed method reduces intensity variation properly, we expect to see the identified cellular population by unsupervised clustering to be similar across the three adjacent sections. We run k-means clustering with $N = 5$, 10, 15, and 20 and calculate correlation coefficient of cell populations as shown in Fig. 5. The top row

illustrates correlation of cell types across three adjacent sections without intensity normalization and bottom row shows correlation of cell types with the intensity normalization approach.

Since the proposed approach reduces intensity variation, the correlation coefficient based on the clustered group component shows high correlation. On the other hand, due to the intensity variation, unsupervised clustering often identifies the same cell types from each section into different groups (i.e., one cluster originates from section 1 and the other from the other section). Thus, without normalization, unsupervised clustering identifies batch effect, i.e., the same cell type can be clustered into different clusters as shown in Supplementary Fig. 1, and the correlation of cell population is lower.

As the number of clusters ($N$) increases, the correlation decreases slightly although the proposed method still improves correlation compared with no normalization. Even though we expect to see similar populations across three adjacent sections, there are still small variations in cell populations, which may separate as the number of clusters increases as shown in Fig. 5 (more details are shown in Supplementary Fig. 1) and cause lower correlation of cell composition. As an example, Supplementary Fig. 1a shows cell population distribution when we cluster cells into five clusters ($N = 5$). Without normalization, clustered ID 1 population is mainly from batch 1 (or section 1), clustered ID 2 population is mainly from batch 2 (or section 2), and clustered ID 4 is mainly from batch 3 (or section 3) due to the intensity variation. On the other hands,

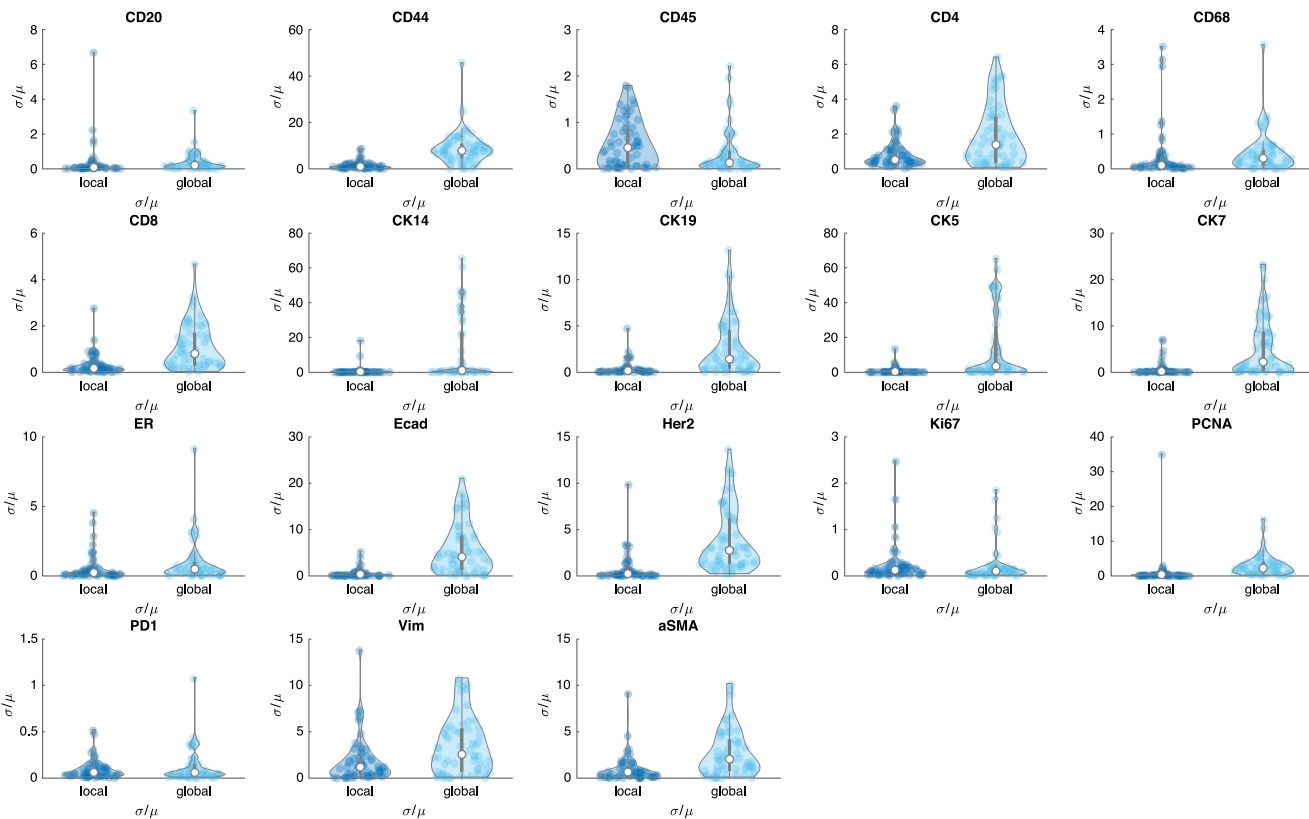

**Fig. 4 Comparison of cofficient of variation across CyCIF markers.** Coefficient of variation ($c_v = \frac{\sigma}{\mu}$) of positive cell counts based on local and global threshold values.

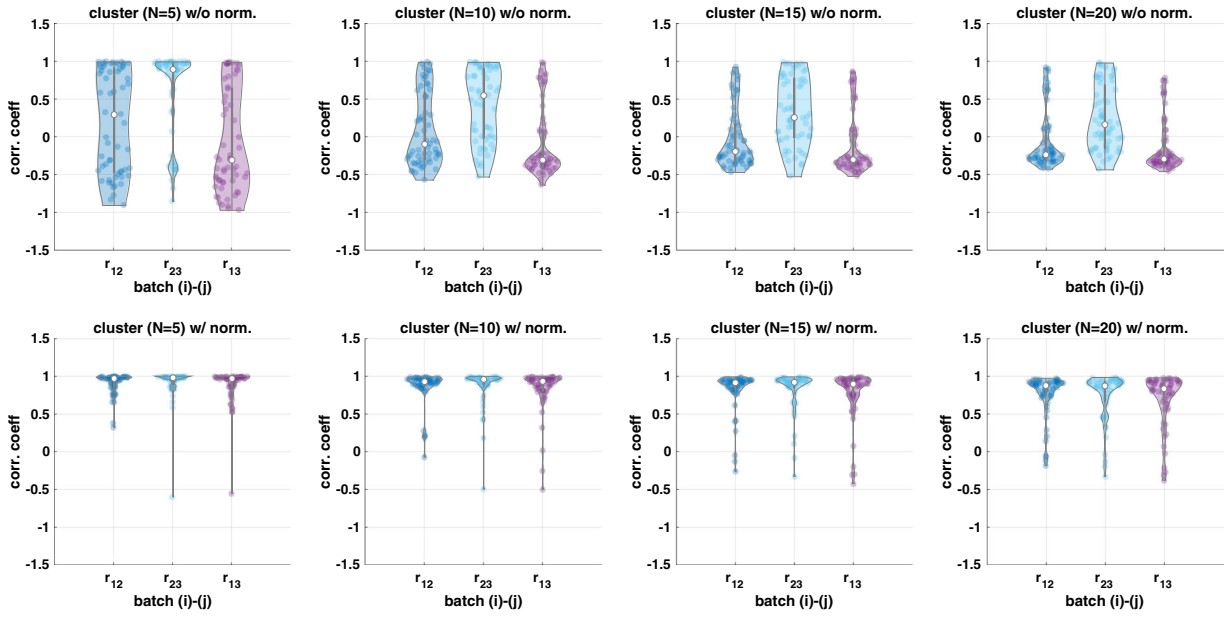

**Fig. 5 Cell population comparison by using unsupervised clustering.** Correlation coefficient based on group component using unsupervised clustering across three adjacent TMA samples ($n = 59$) where top figure shows the result without the proposed approach and bottom figure shows the result with the proposed approach.

with the proposed approach, each clustered ID shows similar distribution across three adjacent sections.

**Application with longitudinal biopsies sample study.** We validate the proposed approach with a longitudinal biopsy sample study, from the Serial Measurements of Molecular and Architectural Responses to Therapy (SMMART) trials[23], where tissue biopsy, fixation, processing, and multiplexed imaging are done at different times. Since we need to identify cellular composition changes for comparative study from longitudinal biopsies (before/after drug treatment), it is critical to remove unwanted variation and integrate two datasets together for

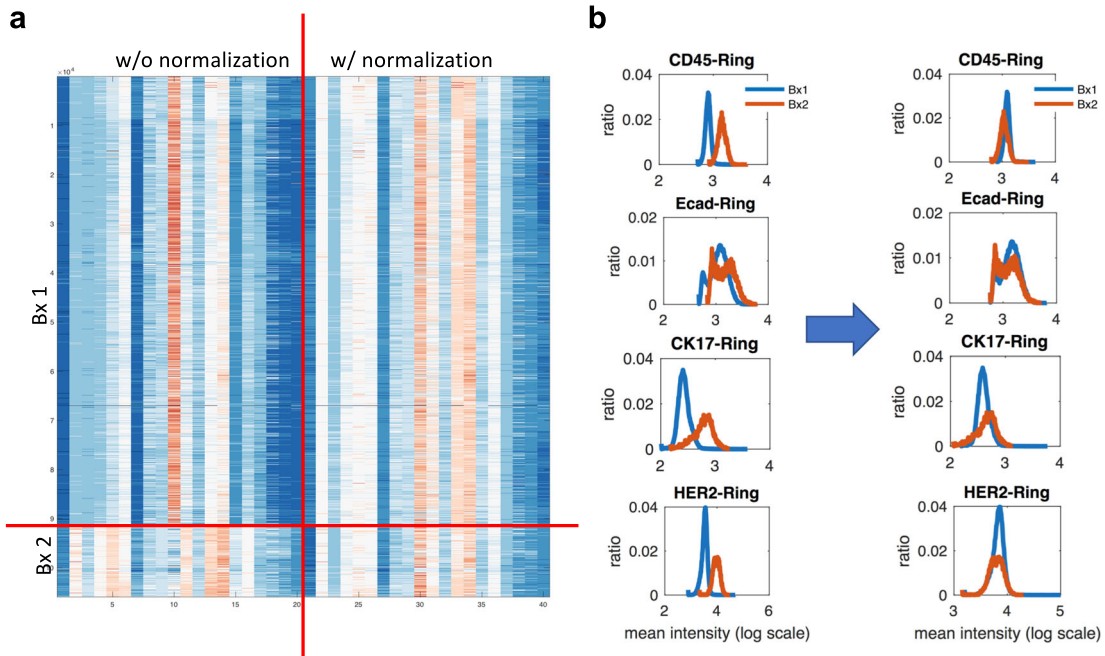

**Fig. 6 Intensity normalization application with longitudinal biopies sample study.** Cyclic multiplexed IF Intensity distribution across Bx 1 and Bx 2: **a** heat map shows intensity features without and with the proposed normalization and **b** intensity distribution of selected markers (CK45, Ecad, CK17, HER2) (all other markers' distribution is shown in Supplementary Fig. 2.

unsupervised clustering analysis. Because of the previous afore-mentioned aim, and because we often observe different cell population in paired biopsies, we cannot use the QN approach.

Figure 6a shows a heat map of intensity profiles. Without normalization, if we apply any unsupervised clustering, it will cluster cells based on their batches (i.e., Bx1 and Bx2) as we can see huge variations in intensity. By applying the proposed approaches, intensity profiles look similar to each other across all the markers. Figure 6b shows intensity distribution changes before/after normalization. As we can see, the proposed approach aligns intensity profiles by removing any unwanted variation and matching the intensity level of negative control. Note that if we use QN instead of using the proposed approach, we may generate an obscure shift of the intensity distribution depending on their cell populations as shown in CK17 and HER2 and we expect that this will result in unwanted artifact. More detailed information such as a comparison between before/after normalization for all other markers is shown in Supplementary Fig. 2.

**Discussion**

In order to infer background signal, for a given $i$th marker, we need to find the most mutually exclusive marker pairs to identify the negative control group. For each marker, we examine all pairs and choose the top 5 mutually exclusive marker pairs based on biologically-known mutually exclusive marker and a data-driven approach (Section: Identifying mutual exclusive maker pairs). Supplementary Table 1 shows a comparison between the data-driven approach (left panel) and biologically-known mutually exclusive marker pairs (right panel). We found that the data-driven approach identifies similar and consistent marker pairs to biologically-known mutually exclusive pairs. The red color in the data-driven approach indicates the matched marker pairs presented in biological knowledge-based approach. Similarly the green color in the biological knowledge-based approach indicates the matched marker pairs presented in the data-driven approach. To apply the proposed approach, there might be potential

challenges such as non-specific staining, inherent variable background signal, variability in mutually exclusive biomarkers, and cell types. We recommend to use both data-driven and biologically-known mutually exclusive marker pairs for validating data for quality control purpose. For instance, if there exists non-specific staining in biologically-known mutually exclusive markers, the marker pair will be positively correlated. In this case, we can use the other mutually exclusive markers to infer background. Herein, as we described, we use biologically-known mutually exclusive markers for inferring autofluorescence signal after visual confirmation. Moreover, one can use more than one mutually exclusive markers to have more robust background inference.

Mutually exclusive marker identification works reasonably well in many cases and confirms that biologically-known mutually exclusive marker pairs show mutual exclusiveness in our CyCIF imaging data. From data-driven approach, in general, immune markers show most mutual exclusiveness with CK markers. Interestingly, we also observe more specific mutual exclusiveness patterns, for instance, the top mutually exclusive pairs with CD20 which is B-cell marker (i.e., lymphoid lineage immune cell with CD45$^+$) are CD68$^+$, a macrophage (myeloid lineage) marker, and CD31$^+$, an endothelial cell marker and it is consistent with biologically-known mutually exclusive cell types.

We observe that a few cells often show stain in both cancer and immune markers and thus, we need to carefully check marker staining quality for this analysis. Herein, we simply use (fixed) biologically-driven mutually exclusive marker pairs to infer background signal, instead of identifying the most mutually exclusive pairs for individual cores from the TMAs dataset and a longitudinal study.

In addition, these potential issues can be addressed mostly in a quality control (QC) step. To support this, we implement a simple visualization tool for multiplexed imaging data based on an open-source platform[26], which will be useful to visually evaluate non-specific staining, inherent variable background signal or mutually exclusive biomarkers as an alternative way to visualize data. As an

example, Supplementary Fig. 3 demonstrates a use-case of a multi-dimensional image viewer with selected markers (CD45 and CK19), which shows mutual exclusiveness as shown in scatter plot.

## Methods

We now propose RESTORE as a practical strategy for batch effect corrections during staining that does not require adding control tissues or using technical replicates in staining and processing. Our approach is composed of two parts: (1) definition of mutually exclusive marker pairs or cell types that are known to be positive for a given markers and negative for others based on biological literature or data-driven based, and (2) inference of background levels for specific markers in cells that are defined to be negative for those markers based on positive identifying markers. The positive/negative marker sets used in this study are defined in Supplementary Table 1.

For a given reference marker, we can use target marker (shown in Supplementary Table 1), which shows mutual exclusiveness or positive-negative associations to define the background levels of negative controls. For a given reference marker, since we expect the background levels of negative controls should be aligned within the same ranges across tissue samples, we can normalize a reference marker expression by the inferred background levels. Also, since we have thousands or millions of cells in tissue and various markers to characterize different cell types, it is not difficult to find a negative control for individual markers from the same tissue sample.

### Concepts for intensity normalization.

Here, we introduce the fundamental concepts for intensity normalization. For a given reference marker, expression levels of the negative control are known a priori to be truly unassociated with the factor of interest. On the other hand, positive control markers are those expression levels that are known a priori to be truly associated with the factor of interest. For example, if the factor of interest is finding immune cells, CD45 would be a positive control, and a negative control would be any cytokeratin (CK5, CK7, CK19, etc.) markers. Since the expression of the negative control group is known to be unassociated with the factor of interest, there is no danger in picking up any of the relevant biology and thus we could use them to remove unwanted variations. Even though individual tissue samples may have different background levels, we identify negative cells for each tissue sample by using mutually exclusive marker pairs in Supplementary Table 1 and then make their expression level below the level of any positive cells by normalizing with the inferred background levels.

For our CyCIF imaging[3], we have tens of markers as shown in Supplementary Table 1. Thus, for a given $i$th marker as a reference, there exists at least one marker, the $j$th marker (target), showing a mutually exclusive expression pattern. The positive cell of the $i$th marker cannot express the positive signal of $j$th marker as shown in Fig. 7a. This is a reasonable assumption; since current CyCIF panel includes immune/cancer markers (biologically mutually exclusive). Therefore, there should exist at least one mutually exclusive marker pair. Herein, we do not consider if there exists no positive cell in the reference marker since this marker is not useful for further analysis by definition (i.e., no positive staining in the tissue sample).

### Identifying mutual exclusive marker pairs.

This procedure can be done by using either (1) biologically-known mutually exclusive marker information (i.e., a cancer vs. immune marker), or (2) a data-driven approach by identifying mutually exclusive information from the $i$th marker and all other maker pairs. For the latter case, we use a singular value decomposition (SVD) to measure the mutual exclusiveness.

Define $D = \begin{bmatrix} x_{i1} & x_{j1} \\ \cdots & \cdots \\ x_{in} & x_{jn} \end{bmatrix} \in \mathbb{R}^{n \times 2}$ where $i, j$ represents the $i$th and $j$th markers, respectively, and $x_{\{\cdot\}}$ represent mean intensity of individual cell. By using SVD, we can factorize $D = U \Sigma V^*$ where $U \in \mathbb{R}^{n \times n}$, $\Sigma \in \mathbb{R}^{n \times 2}$ and $V^* \in \mathbb{R}^{2 \times 2}$. $U$ and $V^*$ can be viewed as rotation matrix and the diagonal entries $\sigma_i$ of $\Sigma$ are known as the singular values of $D$, which can be regarded as a scaling matrix. Since the singular values can be interpreted as the semi-axis of an ellipse in 2D, we measure the mutually exclusive of two marker expressions as a ratio ($r = \frac{\sigma_2}{\sigma_1}$). If two markers are highly correlated with each other, we will get an elongated ellipse (i.e., $r$ is close to zero). On the other hand, if they are mutually exclusive, you will get a more circular shape (i.e., close to one).

### Identifying cell types via non-negative matrix factorization (NNMF) or sparse subspace clustering (SSC).

For a given mutually exclusive marker pair, we define mean intensity profiles of two markers as $\mathcal{Y} = \begin{bmatrix} \mathcal{Y}^1 \\ \mathcal{Y}^2 \\ \mathcal{Y}^3 \end{bmatrix} \in \mathbb{R}^{N \times 2}$, where $N (= \sum_{i=1}^{3} n_i)$ represents the total number of cells, $\mathcal{Y}^1 \in \mathbb{R}^{n_1 \times 2}$ represents 2-dimensional ($i$th and $j$th markers) mean intensity profile of a set of the $i$th marker positive cells, $\mathcal{Y}^2 \in \mathbb{R}^{n_2 \times 2}$ represents intensity profile of a set of the $j$th marker positive cells and $\mathcal{Y}^3 \in \mathbb{R}^{n_3 \times 2}$ represents intensity profile of a set of negative cells for both the $i$th and $j$th marker where $n_i$ represents the number of cells belong to $\mathcal{Y}^i$.

The $l$th row of $\mathcal{Y}$ can be denoted by $y'_l = [y_{l1} \ y_{l2}] = [b_i^l + s_i^l \ b_j^l + s_j^l] \in \mathbb{R}^{1 \times 2}$, where $b_i^l$ and $b_j^l$ represents baseline (autofluorescence level) of the $i$th and $j$th marker, respectively, and $s_i^l$ and $s_j^l$ represents signal level of the $i$th and $j$th marker, respectively. By definition (and expectation of high signal-to-background ratio (SBR) of immunofluorescence intensity profile), we assume $s_i^l \gg b_i^l > 0$ and $s_j^l \gg b_j^l > 0$:

*Lemma 1.* Consider mutually exclusive set $(i, j)$-markers, i.e., $\{\mathcal{Y}^1, \mathcal{Y}^2, \mathcal{Y}^3\}$ where $y'_p \in \mathcal{Y}^p$, $p$ represents group index, i.e., $p = \{1, 2, 3\}$. Then $y_p$ cannot be represented by linear combination of $\alpha y_q + \beta y_r$ with constraint ($\alpha > 0$ and $\beta > 0$) where $p \neq q \neq r$ and $(p, q, r)$ represents any permutations of $(1, 2, 3)$.

*Proof.* (suppose not) i.e., $y_p = \alpha y_q + \beta y_r$ where we simply consider $p = 1$, $q = 2$ and $r = 3$:

$$\begin{bmatrix} b_i^p + s_i^p \\ b_j^p \end{bmatrix} = \alpha \begin{bmatrix} b_i^q \\ b_j^q + s_j^q \end{bmatrix} + \beta \begin{bmatrix} b_i^r \\ b_j^r \end{bmatrix} \quad (1)$$

Then, reformulating this

$$\begin{bmatrix} \alpha \\ \beta \end{bmatrix} = -\frac{1}{s_j^q b_i^r + (b_i^r b_j^q - b_i^q b_j^r)} \begin{bmatrix} b_j^r & -b_i^r \\ -(b_j^q + s_j^q) & b_i^q \end{bmatrix} \begin{bmatrix} b_i^p + s_i^p \\ b_j^p \end{bmatrix} \quad (2)$$

Recall $s_i^l \gg b_i^l$ or simply assume $b_j^r \approx b_j^p \approx b_j^q$, $b_i^r \approx b_i^p \approx b_i^q$:

$$\begin{bmatrix} \alpha \\ \beta \end{bmatrix} = -\frac{1}{s_j^q b_i^r + (b_i^r b_j^q - b_i^q b_j^r)} \begin{bmatrix} (b_j^r b_i^p - b_i^r b_j^p) + b_j^r s_i^p \\ -(b_j^q s_i^p + s_j^q b_i^p + s_j^q s_i^p) + (b_i^q b_j^p - b_i^p b_j^q) \end{bmatrix} \quad (3)$$

$$\approx -\frac{1}{s_j^q b_i^r} \begin{bmatrix} b_j^r s_i^p \\ -(b_j^q s_i^p + s_j^q b_i^p + s_j^q s_i^p) \end{bmatrix} \Rightarrow \begin{bmatrix} (-) \\ (+) \end{bmatrix} \quad (4)$$

(by contradiction). $\square$

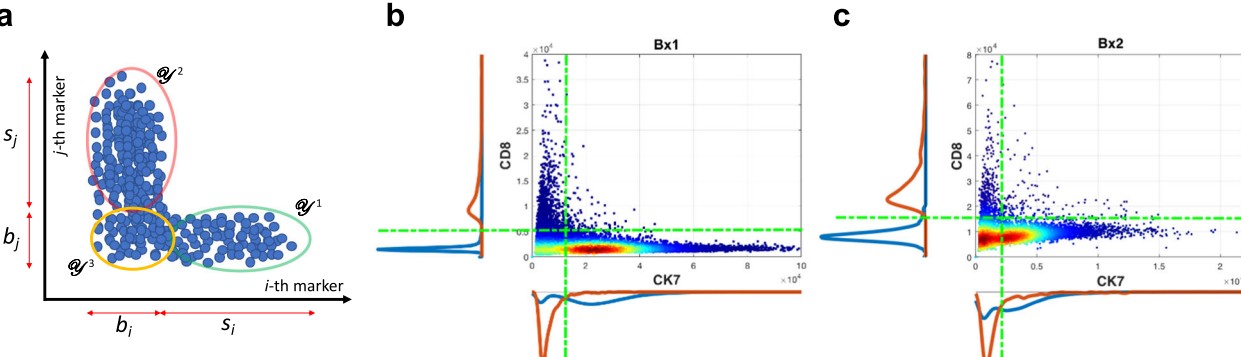

**Fig. 7 A conceptual illustration of inferring intensity level of negative cells based on mutually exclusive marker pairs.** Conceptual illustration (**a**) and example of scatter plot using longitudinal biopsy samples (**b** Bx1 and **c** Bx2): mean intensity distribution of two markers shows different baseline and signal level of each marker (CK7 and CD8) in Bx1/2 where green lines illustrate example of inferred intensity of baseline for each marker, red and blue line shows corresponding cell population density.

By Lemma 1, simple NNMF or SSC or other method (e.g., gaussian mixture model (GMM)) with this constraint should be able to identify the group of cell types for a given mutually exclusive pattern.

**Non-negative matrix factorization (NNMF)**. We can use NNMF:

$$\min \| \mathcal{Y} - WH\|_F \text{ subject to } W \geq 0, H \geq 0 \tag{5}$$

If we consider a 2-dimensional feature set (i.e., mean intensity profile), there exists a trivial solution, i.e., $W = \mathcal{Y}$ and $H = I$ so we need to consider additional constraint such as $H \neq I$. If we include more than 2 dimensional features by adding more features including total intensity, other cellular mean intensity (nuclear-, cytoplasm-, or cellular), we can simply use (5).

**Sparse subspace clustering (SSC)**. We can use SSC with additional constraint by considering that each data point in a union of subspaces can be efficiently reconstructed by a combination of other points in the dataset. More precisely, each data point for data point $y_m = \mathcal{Y}c_m$ where $c_{mm} = 0$ and $c_{ij} \geq 0$ (additional constraint). Then,

$$\min \| c_i \|_q \text{ subject to } y_m = \mathcal{Y}c_m, c_{mm} = 0, c_{ij} \geq 0. \tag{6}$$

$$\min \| \mathbf{C} \|_1 \text{ subject to } \mathcal{Y} = \mathcal{Y}\mathbf{C}, \text{diag}(\mathbf{C}) = 0, c_{ij} \geq 0. \tag{7}$$

In practice, since the intensity values of $y_m$ are all positive and $l_1$ optimization penalizes a sparsity of coefficient $\mathbf{C}$, it does not choose negative coefficient of $c_{ij}$ even without having an additional constraint (i.e., $c_{ij} \geq 0$) in the optimization problem in (6) or (7). In this paper, we simply use the optimization problem in (7) to identify two groups, i.e., positive cell and negative cell group to infer background signal.

**Inferring autofluorescence level with mutually exclusive marker pairs**. Here, we simply use SSC to divide $\mathcal{Y}$ into two groups, either $[\mathcal{Y}^1 \oplus \mathcal{Y}^3$ and $\mathcal{Y}^2]$ or $[\mathcal{Y}^1$ and $\mathcal{Y}^2 \oplus \mathcal{Y}^3]$, by selecting the number of clusters is equal to 2 (note that one could cluster more than 2 and assign each cluster into the group of interest for inferring background or autofluorescence signal). Since we are interested in inferring baseline signal of the $i$th marker, we measure the $i$th marker expression from the clustered group and determine the group for measuring baseline signal:

$$\begin{bmatrix} \mathcal{Y}^1 \\ \mathcal{Y}^2 \\ \mathcal{Y}^3 \end{bmatrix} = \begin{bmatrix} Y_1^1 & Y_2^1 \\ Y_1^2 & Y_2^2 \\ Y_1^3 & Y_2^3 \end{bmatrix} = \begin{bmatrix} B_i^1 + S_i^1 & B_j^1 + S_j^1 \\ B_i^2 + S_i^2 & B_j^2 + S_j^2 \\ B_i^3 + S_i^3 & B_j^3 + S_j^3 \end{bmatrix} = \begin{bmatrix} B_i^1 & B_j^1 \\ B_i^2 & B_j^2 \\ B_i^3 & B_j^3 \end{bmatrix} + \begin{bmatrix} S_i^1 & 0 \\ 0 & S_j^2 \\ 0 & 0 \end{bmatrix} \tag{8}$$

For a given clustered group, either $[\mathcal{Y}^1 \oplus \mathcal{Y}^3$ and $\mathcal{Y}^2]$ or $[\mathcal{Y}^1$ and $\mathcal{Y}^2 \oplus \mathcal{Y}^3]$, we can infer baseline signal of the $i$th marker from either $B_i^2$ (from $\mathcal{Y}^2$) or ($B_i^2$ or $B_i^3$) (from $\mathcal{Y}^2 \oplus \mathcal{Y}^3$). Note that we can measure the autofluorescence level or baseline signal from each clustered group $Y_p^q$ and identify background signal and signal level, respectively, as follows:

$$\mathbb{E}[B_i^2] \approx \mathbb{E}[Y_1^2] \approx \mathbb{E}[B_i^3] \approx \mathbb{E}[Y_1^3] \tag{9}$$

$$\mathbb{E}[B_j^2] \approx \mathbb{E}[Y_2^1] \approx \mathbb{E}[B_j^3] \approx \mathbb{E}[Y_2^3] \tag{10}$$

$$\mathbb{E}[S_i^1] \approx \mathbb{E}[Y_1^1] - \mathbb{E}[Y_1^3] \approx \mathbb{E}[Y_1^1] - \mathbb{E}[Y_1^2] \tag{11}$$

$$\mathbb{E}[S_j^2] \approx \mathbb{E}[Y_2^2] - \mathbb{E}[Y_2^1] \approx \mathbb{E}[Y_2^2] - \mathbb{E}[Y_2^3] \tag{12}$$

**Intensity normalization by inferring autofluorescence signal**. For a given reference marker (i.e., $i$th marker), intensity normalization step is straightforward by the fact that the autofluorescence level of negative controls (the $i$th marker expression of $B_i^1$, $B_i^2$ or $B_i^3$) should be in the same ranges across batches or samples. As we infer background signal based on the negative control, we can scale intensity values by the inferred background signal level of the negative control for individual sample, respectively, to align intensity distribution. One could use the maximum intensity value from the negative control instead of using the mean intensity value. Thus, for a given reference marker, all the background/baseline levels of the negative control are below one, i.e., in the same ranges across samples.

As an example, consider two sample tissues $p$ and $q$ which have different gains $G_p$ and $G_q$. This gain reflects any possible source of intensity variation such as tissue fixation, exposure time, batch effect, etc. Due to the different gain, the intensity values of $p$ and $q$ could be different. For a given $i$th marker, the mean intensity measurement of a single cell can be defined as follows:

$$y_i^p = G_i^p(b_i^p + s_i^p) \triangleq B_i^p + S_i^p$$

$$y_i^q = G_i^q(b_i^q + s_i^q) \triangleq B_i^q + S_i^q$$

where $b_i^p$, $b_i^q$ represent baseline or autofluorescence signal, $s_i^p$, $s_j^q$ represent signal, and $y_i^p$ and $y_j^q$ represent measurement signal, i.e., single-cell mean intensity from

sample $p$ and $q$, respectively. $G_i^p$ and $G_i^q$ represent gain value of the $i$th marker for sample $p$ and $q$, respectively.

We infer $\overline{B}_i^p$ and $\overline{B}_i^q$ using mutually exclusive maker pairs and herein, we assume that we choose $\overline{B}_i^p(= G_i^p\overline{b}_i^p)$ and $\overline{B}_i^q = (G_i^q\overline{b}_i^q)$ from the maximum values from the negative controls (i.e., $\overline{b}_i^p = \max(b_{i,k}^p)$ and $\overline{b}_i^q = \max(b_{i,l}^q)$ where $k$ and $l$ represents cell index for sample $p$ and $q$, respectively). Then, we normalize intensity profiles (i.e., $y_i^p$ and $y_i^q$) based on these values as follows:

$$\overline{y}_i^p \triangleq \frac{y_i^p}{\overline{B}_i^p} = \frac{G_i^p(b_i^p + s_i^p)}{\overline{B}_i^p} = \frac{G_i^p(b_i^p + s_i^p)}{G_i^p\overline{b}_i^p} = \frac{b_i^p}{\overline{b}_i^p} + \frac{s_i^p}{\overline{b}_i^p}$$

$$\overline{y}_i^q \triangleq \frac{y_i^q}{\overline{B}_i^q} = \frac{G_i^q(b_i^q + s_i^q)}{\overline{B}_i^q} = \frac{G_i^q(b_i^q + s_i^q)}{G_i^q\overline{b}_i^q} = \frac{b_i^q}{\overline{b}_i^q} + \frac{s_i^q}{\overline{b}_i^q}$$

Therefore,

$$\overline{y}_i^p = \begin{cases} \frac{b_i^p}{\overline{b}_i^p} (\leq 1), & \text{if negative cell, i.e., } s_i^p = 0 \\ 1 + \frac{s_i^p}{\overline{b}_i^p}, & \text{if positive cell, i.e., } s_i^p > 0 \end{cases},$$

$$\overline{y}_i^q = \begin{cases} \frac{b_i^q}{\overline{b}_i^q} (\leq 1), & \text{if negative cell, i.e., } s_i^q = 0 \\ 1 + \frac{s_i^q}{\overline{b}_i^q}, & \text{if positive cell, i.e., } s_i^q = 0 \end{cases}$$

Note that $\frac{b_i^p}{\overline{b}_i^p} \leq 1$, $\frac{b_i^q}{\overline{b}_i^q} \leq 1$ and $\frac{s_i^p}{\overline{b}_i^p} \gg 1$, $\frac{s_i^q}{\overline{b}_i^q} \gg 1$ by definition (high SBR). So, all the negative cell from the sample $p$ and $q$ will be less than or equal to 1 and positive signal will be above 1 by normalization. Also, note that normalized measurement $\overline{y}_i^p$ and $\overline{y}_i^q$ are not dependent on the gain term $G_i^p$ and $G_i^q$ anymore and if we assume $\overline{b}_i^q \approx \overline{b}_i^q$, normalized signal $\overline{y}_i^p$ and $\overline{y}_i^q$ are representing the true signal, i.e., $s_i$ and $s_j$ with the same scaling factor ($\overline{b}_i^q \approx \overline{b}_i^q$).

It is important to see whether normalization conserves the true signal ratio $\left(\frac{s_i^q}{s_i^p}\right)$ for positive cell. Since normalized intensity for negative cell will be less than or equal to 1, we do not consider here. Assuming $\overline{b}_i^q \approx \overline{b}_i^q = \overline{b}_i$, we consider the signal ratio of two positive cells from $p$ and $q$ samples:

$$\frac{\overline{y}_i^q}{\overline{y}_i^p} = \frac{1 + \frac{s_i^q}{\overline{b}_i}}{1 + \frac{s_i^p}{\overline{b}_i}} = \frac{\overline{b}_i + s_i^q}{\overline{b}_i + s_i^p} \tag{13}$$

If signal level is high enough (high signal-to-baseline ratio (SBR)), i.e., $s_i^p \gg \overline{b}_i$ and $s_i^q \gg \overline{b}_i$,

$$\frac{\overline{y}_i^q}{\overline{y}_i^p} \approx \frac{s_i^q}{s_i^p} \tag{14}$$

Thus, for high SBR region, we could preserve the true signal ratio well but we may have distorted signal ratio when signal level is close to near baseline or background level. On the other hand, without normalization, we assume $\overline{b}_i^p \approx \overline{b}_i^q = \overline{b}_i$ and consider the signal ratio of positive cells from $p$ and $q$ samples as below:

$$\frac{y_i^q}{y_i^p} = \frac{G_i^p(\overline{b}_i + s_i^p)}{G_i^q(\overline{b}_i + s_i^q)} \approx \frac{G_i^p s_i^p}{G_i^q s_i^q} \tag{15}$$

For the last approximated term, we assume that signal level is high enough. We can see that the true signal ratio is scaled by the gain ratio if we do not normalize them. In order to show this, we simulate data and compare the result with and without normalization, i.e., $\frac{y_i^p}{y_i^q}$ and $\frac{\overline{y}_i^p}{\overline{y}_i^q}$. The simulated result and heat map of the result are shown in Supplementary Fig. 4 and 5, respectively. As we described above, as SBR increases, the normalized measurement is close to the true signal ratio. In Supplementary Fig. 5, a heat map shows clearly the region of distortion in low SBR region with high true signal ratio. On the other hand, without normalization, the measurement is converged to gain ratio (i.e., $G_2/G_1$) where we have $G_2/G_1 = 0.5$ in this simulation.

**Statistics and reproducibility**. We evaluate the performance of the proposed method with three adjacent TMA sections ($n = 59$) and results are presented as mean and the full distribution of the sample as specified in the figure. We use the three adjacent TMA sections as technical replicates and compare correlation coefficient of cell population and coefficient of variation as quantitative metrics.

**Ethics**. We purchased TMA tissue sections (BR1506, US Biomax, Derwood, MD). All human tissue is collected under HIPPA approved protocols with the highest ethical standards with the donor being informed completely and with their consent.

**Reporting summary**. Further information on research design is available in the Nature Research Reporting Summary linked to this article.

## Data availability

For research reproducibility, our data (https://www.dropbox.com/sh/mhlep8oashyf2lo/AAAZGva4Cr1pIz0jJHYa9hLXa?dl=0download) is available: (1) 59 TMAs with three adjacent sections and (2) longitudinal biopsies sample.

## Code availability

Our code (in Matlab and Python) is available at https://gitlab.com/Chang_Lab/cycif_int_norm.

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

## Acknowledgements

We thank Ting Zheng and Lydia Grace Campbell for technical assistance, and Lauren Kronebush for helping scientific editing. This work was supported in part by the National Cancer Institute (U54CA209988, U2CCA233280), SBIR (R44CA224994) and Prospect Creek Foundation. Y.H.C. acknowledges the OHSU Center for Spatial Systems Biomedicine, Brenden Colson Center for Pancreatic Care and Biomedical Innovation Program Award from the Oregon Clinical & Translational Research Institute.

## Author contributions

Y.H.C. and J.W.G. conceived and designed the study. Y.H.C. developed algorithms, wrote the code, analyzed the data and interpreted the results. G.T. performed segmentation and feature extraction, E.B. implemented python code with a simple visualization tool for multiplexed imaging data and K.C. optimized the protocol and designed the experiments and J.E. performed the experiments. Y.H.C. drafted the manuscript and K.C., G.T., J.E., E.B., and J.W.G. reviewed the manuscript. J.W.G. supervised and oversaw the study. All authors critically revised the manuscript and provided intellectual content.

## Competing interests

The authors declare no competing interests.
