## [Peer Review File · Communications Biology]

Reviewers' comments:

Reviewer #1 (Remarks to the Author):

The primary goal of this work is to demonstrate a new approach to data normalization for multiplexed immunofluorescence analysis of fixed tissue samples using cells that are purportedly negative for that particular biomarker ("negative control cells") and use that data as background signal. This background data is then used to normalize biomarker signal and reduce between sample variation. The approach was tested and validated using simulated data and 3 sequential slide TMA sections with 59 breast cancer cores that were stained on different days. They also included longitudinal breast biopsy samples that were fixed and processed at different times. Currently, thresholding or gating is the easiest and most commonly used method for delineating positive signal and typically positive or negative controls are not incorporated into the study design. The proposed method presumes that the negative cells are true negatives based on known biology and mutually exclusive marker expression, which is a reasonable starting point. However, non-specific staining and inherent background signal are also present and variable. The authors point out that "As an extreme example, we observed in some cases that the intensity value of negative cells in one tissue were higher than the values of positive cells in a separate sample". The authors use a nice approach to test their approach using three technical replicates and assess the correlations between each slide as a measure of improvement. This is reasonable since in IHC, exact replicates are not possible and 3 sequential sections are as close as one can get. One might still expect to see 10-20% variation between slides, depending on how homogeneous the marker is (greater homogeneity will provide very consistent results, whereas markers that are generally lower frequency will provide more inconsistency). In figure 2, it is unclear what the correlation coefficients represent, there is a wide range from 0.2-1 in the local approach and -0.2-1 in the global approach. Is this a correlation of biomarkers between all the slides?

K-means clustering is used for cell classification and data is shown with and without normalization. Is it necessary to take this step? It is unlikely that the resulting cell populations from clustering (n= 5-15) are distinct cell types. At least within the epithelial cell population, the clusters are likely to represent heterogeneity of expression and cell types. Is it faster/more objective to do K-means clustering on all the data vs separation of immune cell types from epithelial and stroma based on segmentation markers? Based on the biomarkers used, there are 6 "immune markers" (which will have variable expression based on quantity of immune cells in and around the tumor region), 5 cytokeratin markers (prominent in the epithelial cells, but variable), vimentin and SMA (prominent in stromal region), diagnostic markers ER, Her2, Ki67. They identify negative cells for each tissue sample by using mutually exclusive marker pairs. Based on the marker list, there is at least one marker showing a mutually exclusive expression pattern and the top 5 mutually exclusive marker pairs are chosen. The inference of autofluorescence signal is unclear, how accurate is this approach and is there an independent measurement of autofluorescence (e.g. background imaging prior to staining for that round?). Overall, this is an interesting approach and highlights the need for normalization and standardization of data handling. The approach seems a little overcomplicated and I think readers will be interested to know if there is a simpler way to implement. The wider challenge to implement will be in controlling for non-specific staining, inherent variable background signal, variability in mutually exclusive biomarkers and cell types and scaling to whole tissue sections. It would be helpful to explain some of the challenges in more detail.

Minor comments:

- References are incomplete. Gerdes et al (2013) PNAS was the first method to describe multiplexed immunofluorescence (MxIF) using chemical oxidation of dyes and prior to that Schubert et al (2006) Nature Biotechnology, who used light inactivation of dyes.

Reviewer #3 (Remarks to the Author):

In this study, the authors propose a robust intensity normalization method for compensating the unwanted variation in multiplexed imaging. With theoretical simulations and two different datasets, it shows the feasibility of the proposed method for effectively removing the unwanted variation in staining intensity and background fluorescence due to differences in staining between samples, and other artefacts. This is valuable for multiplexed imaging. My main concern is that the proposed method requires a reasonable signal to background ratio. However, the most interesting and critical scenario are the low signal levels (i.e., close to the background signal level). In addition, the authors may consider other criteria of signal-to-noise ratio instead of the signal to background ratio.

RESTORE: Robust intEnSiTy nORmalization mEthod for Multiplexed Imaging COMMSBIO-19-1431-T Authors' response

Dear Associate Editor and Reviewers,

Many thanks for your response and to the reviewers for their helpful and thorough comments. On behalf of the authors, I wish to extend our sincere thanks for your time and effort focused on improving our work. All comments from each of the reviewers are presented below along with our response indicating how we incorporated their feedback into our manuscript. The paper has been much improved as a result. Please find attached an updated manuscript in pdf format with annotated mark-ups of revised changes which are colored in blue.

Comments from Reviewer #1

The primary goal of this work is to demonstrate a new approach to data normalization for multiplexed immunofluorescence analysis of fixed tissue samples using cells that are purportedly negative for that particular biomarker (“negative control cells”) and use that data as background signal. This background data is then used to normalize biomarker signal and reduce between sample variation. The approach was tested and validated using simulated data and 3 sequential slide TMA sections with 59 breast cancer cores that were stained on different days. They also included longitudinal breast biopsy samples that were fixed and processed at different times. Currently, thresholding or gating is the easiest and most commonly used method for delineating positive signal and typically positive or negative controls are not incorporated into the study design. The proposed method presumes that the negative cells are true negatives based on known biology and mutually exclusive marker expression, which is a reasonable starting point. However, non-specific staining and inherent background signal are also present and variable. The authors point out that “As an extreme example, we observed in some cases that the intensity value of negative cells in one tissue were higher than the values of positive cells in a separate sample”.

1. **Comment 1:**

The authors use a nice approach to test their approach using three technical replicates and assess the correlations between each slide as a measure of improvement. This is reasonable since in IHC, exact replicates are not possible and 3 sequential sections are as close as one can get. One might still expect to see 10-20% variation between slides, depending on how homogeneous the marker is (greater homogeneity will provide very consistent results, whereas markers that are generally lower frequency will provide more inconsistency).

Response:

This is a good point. We agree that there may exist small variation between slides so we do not expect perfect correlation coefficient of cell composition across three adjacent sections. In our previous study (Chang, Y.H., Thibault, G., Madin, O., Azimi, V., Meyers, C., Johnson, B., Link, J., Margolin, A. and Gray, J.W., 2017, July. Deep learning based Nucleus Classification in pancreas histological images. In 2017 39th Annual International

Conference of the IEEE Engineering in Medicine and Biology Society (EMBC) (pp. 672-675). IEEE.), we were able to register one section (Hematoxylin and Eosin stain) to the other section (Immunofluorescence imaging) based on nuclei image (H-stain and DAPI) where they are $5\mu m$ difference, which guarantees little variation in cell population within $5\mu m$ difference.

Also, to deal with variation of specific cell (or marker) variation, we show the result with various metrics: A) compare correlation coefficient based on the overall population in Figure 2, B) correlation of coefficient based on cell component across individual markers in Figure 3 and C) the coefficient of variation in Figure 4. As the reviewer pointed out (the result may depend on homogeneous marker expression, frequency of positive cell, etc), global approach (without the proposed approach) shows different variation across markers (Figure 4, coefficient of variation of positive cell count) but the proposed approach minimize these variation across markers.

We added this: *Note that in our previous study, we were able to register one section (Hematoxylin and Eosin stain) to the other section (IF imaging) based on nuclei staining (H-stain and DAPI) where they are $5\mu m$ thickness difference, which guarantees little variation in cell population within $5\mu m$ difference.*

2. Comment 2:

In figure 2, it is unclear what are the correlation coefficients represent, there is a wide range from 0.2-1 in the local approach and -0.2-1 in the global approach. Is this a correlation of biomarkers between all the slides?

Response:

Thank you for raising this point. We regret that our description did not sufficiently clarify the correlation coefficient here. It is “correlation coefficient of cell composition”. In Figure 2, each dot represent single core in TMA. As we described, since each TMA sample has three adjacent tissue samples, we could consider them as almost technical replicates where we expect to see a similar cell type component in their population. For instance, if there is no intensity variation across these three adjacent tissue samples, no matter where we draw threshold lines or gates, the cell populations should be similar (high correlation coefficient of cell composition). Using our approach, we have high correlation coefficient of cell composition but without the proposed approach, correlation coefficient of cell composition varies -0.2 to 1. We also noted that in Figure 2 (middle, the proposed result), three TMAs cores show poor correlation (below 0.8) coefficient but we confirm that those three cores show technical artifacts such as segmentation or tissue loss.

We added this in Figure 2: *where each dot represents single TMA core. Note that in the result of the proposed approach, three TMA cores show poor correlation coefficient (below 0.8) but we confirm that those three cores show technical artifacts such as segmentation or tissue loss.*

3. Comment 3:

K-means clustering is used for cell classification and data is shown with and without normalization. Is it necessary to take this step? It is unlikely that the resulting cell populations from clustering ($n= 5-15$) are distinct cell types. At least within the epithelial cell population, the clusters are likely to represent heterogeneity of expression and cell types. Based on the biomarkers used, there are 6 “immune markers” (which will have variable expression

based on quantity of immune cells in and around the tumor region), 5 cytokeratin markers (prominent in the epithelial cells, but variable), vimentin and SMA (prominent in stromal region), diagnostic markers ER, Her2, Ki67. Is it faster/more objective to do K-means clustering on all the data vs separation of immune cell types from epithelial and stroma based on simple segmentation markers?

Response:

These are fair questions. As we pointed out, we define cell population by two approaches: 1) counting positive cells for individual markers by inferring background signal and 2) using an unsupervised clustering approach as many research groups use for cell type identification of multiplexed imaging data. We do agree that the resulting cell populations from clustering may not show distinct cell types so we vary the number of clusters to capture various cell types.

More importantly, we use unsupervised clustering approach to define cell subpopulation across three adjacent sections. As we mentioned earlier, we consider three adjacent tissue samples as almost technical replicates and expect to see a similar cell subpopulation if there is no intensity variation. As we showed in Figure S1, population distribution of clustering analysis with the proposed intensity normalization approach shows equal distribution of cell type across three adjacent sections. However, without using normalization, due to the intensity variation, some cluster only contain one TMA section. In this case, clustering approach identifies almost batch effect. This highlighted the need for intensity normalization and standardization of data handling and demonstrated robust performance.

We revised this: Since the proposed approach reduces intensity variation, the correlation coefficient based on clustered group component shows high correlation. On the other hand, due to the intensity variation, unsupervised clustering often identifies the same cell types from each section into different groups (i.e., one cluster originates from section 1 and the other from the other section). Thus, without normalization, unsupervised clustering identifies almost batch effect, i.e., the same cell type can be clustered into different clusters as shown in Figure S1, and the correlation of cell population is lower.

4. Comment 4:

They identify negative cells for each tissue sample by using mutually exclusive marker pairs. Based on the marker list, there is at least one marker showing a mutually exclusive expression pattern and the top 5 mutually exclusive marker pairs are chosen. The inference of autofluorescence signal is unclear, how accurate is this approach and is there an independent measurement of autofluorescence (e.g. background imaging prior to staining for that round?). Overall, this is an interesting approach and highlights the need for normalization and standardization of data handling. The approach is a little overcomplicated and I think readers will be interested to know if there is a simpler way to implement. The wider challenge to implement will be in controlling for non-specific staining, inherent variable background signal, variability in mutually exclusive biomarkers and cell types and scaling to whole tissue sections. It would be helpful to explain some of the challenges in more detail.

Response:

Thank you for raising this point. In Table S1, we showed top 5 marker pairs to validate data-driven approach identifies biologically known mutually exclusive marker pairs, vice versa. As we showed in Figure 7, for a given i -th marker (reference), the inference of aut-

ofluorescence signal is based on the mutually exclusive marker expression pattern. Since they are biologically mutually exclusive, each cell cannot present positive intensity of both markers.

First of all, we shared our code (https://gitlab.com/Chang_Lab/cycif_int_norm) in Github so readers do not need to implement by themselves. They can simply run our code. Also, although we proposed either NNMF and SSC approaches here, as we mentioned, simple approach such as Gaussian Mixture Model (GMM) can be applicable. Second, as the reviewer's pointed out, if there are non-specific staining, inherent variable background signal, or variability in mutually exclusive biomarker, we recommend to use both data-driven and biologically known mutually exclusive marker pairs for validating data as quality control purpose. For instance, if there exists non-specific staining in known biologically mutually exclusive markers, it will show correlated population. In this case, we can use the other mutually exclusive markers to infer background. Herein, as we described, we use biologically known mutually exclusive markers for inferring autofluorescence signal after visual confirmation. A user can use more than one mutually exclusive markers to have more robust background inference.

In addition, we implement a simple visualization tool for multiplexed imaging data based on open source platform (<https://github.com/napari/napari>) which will be useful to visually evaluate non-specific staining, inherent variable background signal, variability in mutually exclusive biomarkers as alternative way to visualize data instead of using Figure 7. In Figure S3 (added), we demonstrate that multi-dimensional image viewer with selected markers (CD45 and CK19) can show mutually exclusiveness in parallel to scatter plot, and identify potential issues on non-specific staining, inherent variable background signal, and variability in mutually exclusive biomarkers visually.

We revised: By Lemma 1, simple non-negative matrix factorization (NNMF) or sparse subspace clustering (SSC) or other method (e.g. gaussian mixture model (GMM)) with this constraint should be able to identify the group of cell types for a given mutually exclusive pattern.

To apply the proposed approach, there might be potential challenges such as non-specific staining, inherent variable background signal, variability in mutually exclusive biomarkers and cell types. We recommend to use both data-driven and biologically known mutually exclusive marker pairs for validating data as quality control purpose. For instance, if there exists non-specific staining in known biologically mutually exclusive markers, it will show positive correlation. In this case, we can use the other mutually exclusive markers to infer background. Herein, as we described, we use biologically known mutually exclusive markers for inferring autofluorescence signal after visual confirmation. Moreover, one can use more than one mutually exclusive markers to have more robust background inference.

In addition, these potential issues can be addressed mostly in quality control (QC) step. To support this, we implement a simple visualization tool for multiplexed imaging data based on open source platform [24] which will be useful to visually evaluate non-specific staining, inherent variable background signal or mutually exclusive biomarkers as an alternative way to visualize data. As an example, Figure S3 demonstrates use-case of multi-dimensional image viewer with selected markers (CD45 and CK19) which shows mutually exclusiveness

as shown in scatter plot (right).

5. Comment 6:

Minor comments: - References are incomplete. Gerdes et al (2013) PNAS was the first method to describe multiplexed immunofluorescence (MxIF) using chemical oxidation of dyes and prior to that Schubert et al (2006) Nature Biotechnology, who used light inactivation of dyes.

Response:

This is a good suggestion, and has been addressed to include the author's recommended citations, among others.

We added: Gerdes, M.J., Sevinsky, C.J., Sood, A., Adak, S., Bello, M.O., Bordwell, A., Can, A., Corwin, A., Dinn, S., Filkins, R.J. and Hollman, D., 2013. Highly multiplexed single-cell analysis of formalin-fixed, paraffin-embedded cancer tissue. *Proceedings of the National Academy of Sciences*, 110(29), pp.11982-11987.

Schubert, W., Bonnekoh, B., Pommer, A.J., Philipsen, L., Bckelmann, R., Malykh, Y., Gollnick, H., Friedenberger, M., Bode, M. and Dress, A.W., 2006. Analyzing proteome topology and function by automated multidimensional fluorescence microscopy. *Nature biotechnology*, 24(10), p.1270.

Comments from Reviewer #3

In this study, the authors propose a robust intensity normalization method for compensating the unwanted variation in multiplexed imaging. With theoretical simulations and two different datasets, it shows the feasibility of the proposed method for effectively removing the unwanted variation in staining intensity and background fluorescence due to differences in staining between samples, and other artefacts. This is valuable for multiplexed imaging.

1. Comment 1:

My main concern is that the proposed method requires a reasonable signal to background ratio. However, the most interesting and critical scenario are the low signal levels (i.e., close to the background signal level).

Response:

This is a good point. We believe that we addressed this scenario in our simulation study. In section 4.5., we had simulation study to demonstrate whether the original signal ratio is conserved or not. In Figure S4, from left to right across columns, signal to background ratio (SBR) increases and from top to bottom rows, ground truth signal ratio increases. As we showed Abs. error (w/norm) in bottom right heatmap, signal distortion only exists when signal level is close to near baseline or background level and ground truth signal ratio is high. If we have reasonable signal to background ratio in multiplexed imaging marker, we do not see any distorted signal ratio. However, without the proposed approach (bottom middle in Figure S4), the signal ratio from the measurement is distorted and converged to gain ratio.

Note that the proposed method is focused on dealing with intensity variation, not improving low signal levels. We believe that the low signal levels should be addressed in the development of immunoassays or optimization of antibodies. In section 4.1, we mentioned

that “Herein, we do not consider if there exists no positive cell in the reference marker since this marker is not useful for further analysis by definition (i.e., no positive staining in the tissue sample)”. With a similar reason, if some multiplexed imaging markers show the low signal levels, we would like to optimize staining protocol itself or change antibodies to improve signal-to-background level, rather than computationally improving low signal levels.

2. **Comment 2:**

In addition, the authors may consider other criteria of signal-to-noise ratio instead of the signal to background ratio.

Response:

We regret that our description was unclear. Herein, for a given marker, we consider intensity level of negative cells (mutually exclusive marker) as noise or background interchangeably. Similarly, for a given marker, background is referring intensity level of negative cells so it could represents autofluorescence level. As an example, if we look into immune cell marker (CD45), the most mutually exclusive markers are cancer markers (CKs-). Then, for a given reference marker (CD45), intensity level of CD45 expression of cancer cells would be background level of CD45 expression by our definition.

We added: We now propose **RESTORE** for image quantification in a multiplexed imaging platform to address these issues. A key feature in **RESTORE** is the recognition that some cell types defined by reference markers in the multiplex can be safely assumed to have background or noise levels¹ for other markers in the multiplex. For example, immune cells can be assumed to have background or noise levels

We again wish to sincerely thank the reviewers for their many helpful and insightful comments. On behalf of the authors, we hope the above modifications make our manuscript suitable for publication in the Communications Biology.

Sincerely,
Young Hwan Chang

¹Herein, we consider intensity level of negative cells defined by mutually exclusive marker pairs as background or simply noise level (interchangeably).

REVIEWERS' COMMENTS:

Reviewer #1 (Remarks to the Author):

The changes made are acceptable and improve the quality and value of the paper.

Reviewer #3 (Remarks to the Author):

The authors addressed my concerns.